# Isolated abnormal diffusing capacity for carbon monoxide (iso↓DLco) is associated with increased respiratory symptom burden in people with HIV infection

Katerina L. Byanova[1]*, Jessica Fitzpatrick[2], Amanda K. Jan[2], Maggie McGing[2], Marlena Hartman-Filson[2], Carly K. Farr[3], Michelle Zhang[2], Kendall Gardner[2], Jake Branchini[2], Robert Kerruish[4], Sharvari Bhide[2], Aryana Bates[2], Jenny Hsieh[5], Rebecca Abelman[2], Peter W. Hunt[6], Richard J. Wang[1], Kristina A. Crothers[3,7], Laurence Huang[1,2]

1 Department of Medicine, Division of Pulmonary, Critical Care, Allergy and Sleep Medicine, University of California San Francisco, San Francisco, California, United States of America, 2 Department of Medicine, Division of HIV, Infectious Diseases, and Global Medicine, University of California San Francisco, San Francisco, California, United States of America, 3 Department of Medicine, Division of Pulmonary, Critical Care and Sleep Medicine, Veterans Affairs Puget Sound Health Care System, Seattle, Washington, United States of America, 4 School of Medicine, Western Michigan University Homer Stryker MD School of Medicine, Kalamazoo, Michigan, United States of America, 5 Department of Anesthesia and Perioperative Care, Division of Respiratory Care Services, Zuckerberg San Francisco General Hospital, San Francisco, California, United States of America, 6 Department of Medicine, Division of Experimental Medicine, University of California San Francisco, San Francisco, California, United States of America, 7 Department of Medicine, Division of Pulmonary, Critical Care and Sleep Medicine, University of Washington, Seattle, Washington, United States of America

* katerina.byanova@ucsf.edu

**Data Availability Statement:** All relevant data are within the paper and its Supporting Information files.

## Abstract

### Objectives

An isolated reduction in the diffusing capacity for carbon monoxide (DLco; iso↓DLco) is one of the most common pulmonary function test (PFT) abnormalities in people living with HIV (PWH), but its clinical implications are incompletely understood. In this study, we explored whether iso↓DLco in PWH is associated with a greater respiratory symptom burden.

### Study design

Cross-sectional analysis

### Methods

We used ATS/ERS compliant PFTs from PWH with normal spirometry (post-bronchodilator FEV1/FVC ≥0.7; FEV1, FVC ≥80% predicted) from the I AM OLD cohort in San Francisco, CA and Seattle, WA, grouped by DLco categorized as normal (DLco ≥lower limit of normal, LLN), mild iso↓DLco (LLN >DLco >60% predicted), and moderate-severe iso↓DLco (DLco ≤60% predicted). We performed multivariable analyses to test for associations between DLco and validated symptom-severity and quality of life questionnaires, including the

**Funding:** This study is funded by NIH R01 HL128156, R01 HL128156-06S1, R01 HL090335, R01 HL143998 (all four to LH). RJW was supported by K23 HL162593 and K12 HL143961. KLB was supported by F32 HL166065. PH has a research grant from Gilead; he has received honoraria for lectures from Viiv and Gilead and has consulted for Viiv and Biotron. He is the also the chair for a NIH-funded clinical trial of a drug donated by Merck. The funders had no role in study design, data collection and analysis, decision to publish, or preparation of the manuscript.

**Competing interests:** The authors have declared that no competing interests exist.

modified Medical Research Council dyspnea scale (mMRC), the COPD Assessment Test (CAT), and St. George's Respiratory Questionnaire (SGRQ), as well as between DLco and individual CAT symptoms.

## Results

Mild iso↓DLco was associated only with a significantly higher SGRQ score. Moderate-severe iso↓DLco was associated with significantly higher odds of mMRC ≥2 and significantly higher CAT and SGRQ scores. PWH with moderate-severe iso↓DLco had increased odds of breathlessness, decreased activity, lower confidence leaving home, and less energy.

## Conclusions

Iso↓DLco is associated with worse respiratory symptom scores, and this association becomes stronger with worsening DLco, suggesting that impaired gas exchange alone has a significant negative impact on the quality of life in PWH. Additional studies are ongoing to understand the etiology of this finding and design appropriate interventions.

## Introduction

People living with HIV (PWH) have an increased burden of respiratory symptoms such as cough, phlegm, and breathlessness compared to HIV-negative individuals, independent of viral suppression and antiretroviral therapy (ART) compliance [1–4]. Respiratory symptoms are more severe in PWH who actively smoke or who have emphysema on imaging, obstructive lung disease based on an abnormal ratio of forced expiratory volume in one second/forced vital capacity (FEV1/FVC), or an abnormal diffusing capacity for carbon monoxide (DLco) [4–7].

A reduced DLco is the most common abnormality on pulmonary function testing (PFTs) in PWH [4, 8, 9]. DLco abnormalities are most frequently studied in people with concomitant abnormal spirometry, suggestive of underlying obstructive or restrictive lung disease. However, the most frequent PFT finding in PWH is DLco impairment with normal spirometry, or isolated abnormal DLco (iso↓DLco), which has a prevalence of 16–56% [4, 10]. Estimates of the prevalence of iso↓DLco in HIV-negative individuals in the general population varies from 0.5–12% [11, 12], increasing to 30% as a result of occupational exposures [13]. One reference specifically noted that in their single-center experience, iso↓DLco was a result of pseudonormalization of spirometry due to the presence of combined emphysema and restrictive lung disease [12]. Importantly, abnormal DLco is associated with increased mortality independent of spirometric findings both in PWH [14] and in the general population [15, 16].

DLco has been studied extensively in HIV-negative individuals who smoke but do not have known chronic obstructive pulmonary disease (COPD). Smokers without COPD have worse quality of life and more respiratory symptoms for every 10% predicted decline in their DLco regardless of spirometric findings [17]. Abnormal DLco in smokers with normal spirometry and no clinical or radiographic evidence of lung disease carries a significantly higher risk of developing COPD [18]. Additionally, former smokers with iso↓DLco have been shown to have early emphysema on 3He MRI imaging even with normal chest CT scans [19]. These data suggest that in people who smoke, DLco may be more sensitive than spirometry in identifying

early emphysema and risk of progression to COPD, and that DLco is associated with worse quality of life even in the absence of spirometric obstruction.

In PWH, HIV is associated with a lower DLco even in the absence of emphysema [20]. Abnormal DLco in PWH has not been studied specifically in those with normal spirometry, and thus the mechanisms driving iso↓DLco and its clinical sequelae are not well-understood in PWH despite the high prevalence of this PFT finding. PWH with abnormal DLco (regardless of spirometry) have higher IL-6 levels compared to PWH with normal DLco and to HIV-negative individuals [21]. Additionally, PWH with iso↓DLco have a unique inflammatory marker pattern compared to those with both abnormal DLco and spirometry [10], suggesting that the etiology of iso↓DLco may differ based on HIV status and may be independent from the pathways driving COPD in PWH.

Whether iso↓DLco among PWH is associated with any respiratory symptoms or whether it is simply a marker of future complications is not currently known. Only one prior study has specifically explored symptoms associated with abnormal DLco in PWH (with any spirometry), and it showed an increased prevalence of cough, phlegm and dyspnea in PWH with DLco <60% predicted (a moderate-severe reduction in DLco) compared to PWH with DLco ≥60% and HIV-negative controls [4].

In this cross-sectional analysis, we sought to determine whether the iso↓DLco lung function pattern was associated with clinically relevant changes in respiratory symptoms and quality of life as measured by the mMRC dyspnea scale, COPD Assessment Test (CAT), and St. George's Respiratory Questionnaire (SGRQ), as well as with individual respiratory symptoms. Identifying such associations would be a key step in validating the clinical importance of this PFT finding and would also highlight the need for further studies to understand its etiology.

## Materials and methods

### Study cohort

Inflammation, Aging, Microbes, and Obstructive Lung Disease (I AM OLD) Study: I AM OLD is a longitudinal cohort of PWH in the United States (San Francisco, CA and Seattle, WA) and Kampala, Uganda assessed for incidence and progression of lung function abnormalities over time. In the US, I AM OLD is comprised of adult PWH (age ≥18) recruited at the Zuckerberg San Francisco General Hospital and Seattle's Harborview Medical Center. Participants are enrolled during routine outpatient HIV clinic visits or during hospitalizations for acute pneumonia. Study visits occur annually, and during each visit without evidence of acute pneumonia, participants undergo blood draws, PFTs, and symptom assessment. For individuals enrolled during a pneumonia hospitalization, baseline PFTs are conducted three months after completion of treatment, and only if the acute symptoms have resolved. Participants provide written informed consent. The study protocol is approved by the Institutional Review Boards of the University of California San Francisco (IRB#13–11328) and the University of Washington (IRB#STUDY00002542).

### Study design

This is a cross-sectional analysis of US-based I AM OLD participants with normal spirometry as defined by a post-bronchodilator (post-BD) FEV1/FVC≥0.70, post-BD FEV ≥80% predicted and post-BD FVC≥80% predicted [22]. We included all participants with normal spirometry that met American Thoracic Society/European Respiratory Society (ATS/ERS) criteria for acceptability and reproducibility with grades A, B, or C and with at least one acceptable DLco measurement [23, 24]. For individuals with more than one PFT that fit the criteria, the earliest time point was used.

Our primary predictor was DLco impairment. Our primary outcomes were patient-reported respiratory symptom scores based on the mMRC dyspnea scale, CAT [25], or SGRQ [26]. As secondary analyses, we examined the association of DLco impairment with each individual CAT question, as well as with scores from the three main SGRQ categories: symptoms, activity, and impact. We chose to use mMRC because it is a long-established tool for classifying the severity of dyspnea, the symptom we expected DLco to be most significantly associated with, across a range of lung conditions [27–29]. Although our patients by definition do not have COPD as defined by abnormal spirometry, we used CAT as it offered a validated method for evaluating common respiratory symptoms. Finally, SGRQ was chosen for its broad validity across several respiratory conditions in measuring respiratory-related quality of life [26].

## Data collection

Study personnel obtained demographic data, smoking and illicit drug use history, ART use, and history of prior pulmonary illnesses using a standardized questionnaire on the same day as blood draws and PFTs. We measured HIV RNA levels, CD4 counts, and CD8 counts. PFTs consisted of pre- and post-bronchodilator spirometry and DLco measurements. Bronchodilation was achieved by administering 360μg albuterol via a metered-dose inhaler with a spacer. Diffusing capacity was corrected for same-day hemoglobin and carboxyhemoglobin. PFTs were performed in accordance with ATS/ERS guidelines by trained technicians [23, 24]. A single trained reader overread the PFTs to ensure compliance with ATS/ERS standards. To calculate percent predicted values, we used the NHANES III reference equations for spirometry and NHANES I reference equations for DLco [30, 31].

## Statistical analysis

Participants were grouped by corrected DLco values into normal DLco (DLco ≥lower limit of normal, LLN), mild iso↓DLco (LLN >DLco >60% predicted), and moderate-severe iso↓DLco (DLco ≤60% predicted) [4, 7, 32]. Demographic, clinical, and laboratory characteristics, as well as symptom questionnaires and PFTs, were summarized using counts with percent of total (%) for categorical variables, medians with interquartile ranges (IQRs) for skewed continuous variables, and means with standard deviations (SD) for normally distributed continuous variables. Continuous variables were also categorized using clinically relevant cutoffs (e.g., HIV viral load detectable when >40 copies/mL). We examined the associations of *a priori* selected clinical factors and symptom scores in bivariate analyses using separate generalized linear models. Factors known to be related to lung disease, such as cigarette smoking, and ones that were significant at $P<0.2$ in bivariate analyses were included in the multivariable analyses. We performed multivariable analyses using gamma Generalized Linear Models (GLM) with log link to test for associations between DLco and CAT/SGRQ scores. This model was chosen to account for the non-linearity of the outcome variables (symptom scores). We adjusted the GLM models for age, sex, BMI, smoking status (ever smoker yes/no), and history of bacterial pneumonia; SGRQ analyses were additionally adjusted for site location (San Francisco or Seattle) and history of injection drug use based on bivariate analyses results. Multivariable analyses using logistic regression were used to test for associations between iso↓DLco and mMRC scores, grouped into mMRC <2 (normal/mild dyspnea) and mMRC ≥2 (moderate/severe dyspnea), with age, gender, BMI, smoking status (ever smoker yes/no), bacterial pneumonia, and site location (San Francisco or Seattle) as covariates. We also repeated our multivariable analyses looking for associations between DLco and mMRC, CAT, SGRQ, and individual CAT questions with DLco as a continuous variable using logistic regressions and log-gamma GLM.

We tested for associations between DLco and CAT symptoms using a nonparametric Mantel-Haenszel Test for trend. Multivariable analyses using logistic regressions tested the individual associations between DLco categories and CAT symptoms using the same covariates used for total symptom scores. The Kruskal-Wallis test was performed to test for associations between DLco and SGRQ category scores due to the skewed nature of the SGRQ scores.

Data were analyzed with R Studio, version 4.1.1, August 2021, Boston, MA.

## Results

### Cohort

I AM OLD enrolled 316 participants between April 2013 and June 2022 (Fig 1). Of those, 10 were excluded for missing data (one did not have measured DLco, and nine were missing questionnaire data), 13 had spirometry that did not meet ATS/ERS criteria, and 104 were excluded from this analysis for abnormal spirometry (62 had post-BD FEV1/FVC <0.70, and 42 had both post-BD FEV1 and FVC <80% predicted). Therefore, we included a total of 189 participants with normal spirometry in this study.

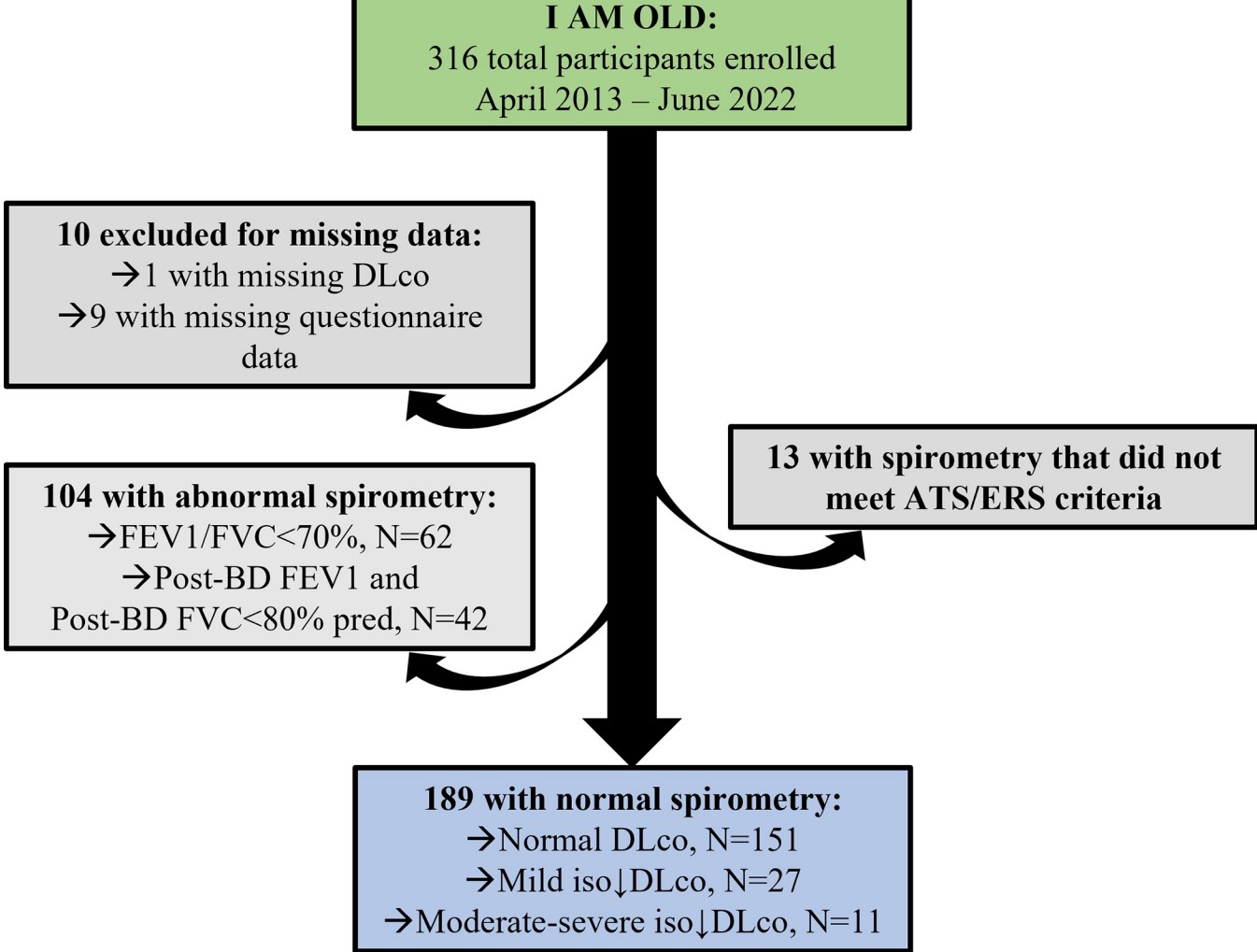

**Fig 1. Participant selection from the I AM OLD cohort.**

## Participants

Of the 189 participants with normal spirometry, 151 (79.9%) had normal DLco, 27 (14.3%) had mild iso↓DLco, and 11 (5.8%) had moderate-severe iso↓DLco (Fig 1 & Table 1). The participants' median age was 50 years (Table 1). One hundred and sixty (85%) were male and 29 (15%) were female. Overall, 83 individuals (44%) were white, 59 (31%) were African American, 30 (16%) were Hispanic, and 17 (9%) identified as 'Other.' One hundred and twenty-two (65%) were ever-smokers, and 73 (39%) had used injection drugs. One hundred forty-three out of 163 (88%) were currently on ART; viral load was undetectable in 131 (72%), and the median CD4 count in the cohort was 509 cells/μl. Eighty-eight participants (48%) had a history of bacterial pneumonia, 43 (23%) had a history of *Pneumocystis* pneumonia (PCP), and 13 (7%) had a history of tuberculosis.

Participants in the three groups differed by age (mild iso↓DLco being the youngest, moderate-severe iso↓DLco the oldest, $P = 0.044$), sex (higher proportion of women in moderate-severe iso↓DLco group, $P = 0.005$), race/ethnicity ($P = 0.021$), tobacco use (highest in mild iso-↓DLco, followed by moderate-severe iso↓DLco, $P = 0.039$), BMI (BMI decreased with worsening DLco, $P = 0.010$), viral load ($P = 0.022$), and history of PCP (highest in moderate-severe iso↓DLco, $P<0.001$). History of bacterial pneumonia also was more prevalent among participants with more severe DLco impairment, although the trend did not reach statistical significance (p = 0.059). Groups were comparable by all other demographic and clinical characteristics.

Participants in each group also differed by their pulmonary function data. While remaining within the normal range, post-BD FEV1 (in liters) and post-BD FVC (in liters) were significantly lower in the moderate-severe iso↓DLco group. The differences in post-BD FVC% predicted and the post-BD FEV1/FVC ratio were not significant. Median DLco was 26.0, 22.6, and 13.7 ml/min/mmHg for the normal, mild iso↓DLco, and moderate-severe iso↓DLco groups, respectively, with median % predicted of 84%, 65.7%, and 54.9%, respectively.

## Iso↓DLco and respiratory symptom scores

The median mMRC scores were 1 (IQR 0, 1), 0 (IQR 0, 1.5) and 3 (IQR 1, 3), in the normal DLco, mild iso↓DLco, and moderate-severe iso↓DLco groups, respectively ($P = 0.007$; Table 1). Across the same three groups, the median CAT scores were 7 (IQR 2, 12), 11 (IQR 6, 17), and 18 (IQR 10, 24), respectively ($P = 0.002$), and the median SGRQ scores were 10 (IQR 3, 25), 20 (IQR 11, 28) and 51 (IQR 23, 61), respectively (P<0.001). Fig 2 is a graphical representation of the unadjusted symptom scores by respiratory questionnaire and DLco category.

We performed multivariable analyses using logistic regressions to test for associations between iso↓DLco and mMRC. The odds of having an mMRC ≥2 were not significantly increased for participants with mild iso↓DLco (aOR 1.58, 95% CI 0.54, 4.64; $P = 0.40$) compared to normal DLco but were 7.03 times higher in PWH with moderate-severe iso↓DLco (95% CI 1.73, 28.5; $P = 0.006$; Table 2). Multivariable analyses using log-gamma GLM showed that while CAT scores were not significantly increased with mild iso↓DLco (2.85-point difference, 95% CI -0.18, 5.89; $P = 0.065$), they were significantly higher in moderate-severe iso-↓DLco (9.44-point difference, 95% CI 2.34, 16.5; $P = 0.009$) compared to the scores of those with normal DLco. Multivariable analyses of DLco and SGRQ scores showed significantly higher SGRQ scores in the mild iso↓DLco (9.29-point difference, 95% CI 0.04, 18.5; $P = 0.049$) and in the moderate-severe iso↓DLco group (26.7-point difference, 95% CI 4.96, 48.4; $P = 0.016$), compared to the normal DLco group. The differences in CAT and SGRQ scores between groups were consistently higher than the minimal clinically important difference (MCID) scores reported in the literature (two and four points, respectively) [33, 34]. The

**Table 1. Participant baseline demographic and clinical characteristics (N = 189).**

| Variable | N | Overall (N = 189) | Normal DLco (% predicted ≥ LLN) (N = 151) | Mild iso↓DLco (%predicted > 60% & < LLN) (N = 27) | Moderate-severe iso↓DLco (% predicted ≤ 60%) (N = 11) | P value |
|---|---|---|---|---|---|---|
| **Demographic characteristics** | | | | | | |
| Age | 189 | 50 (39, 58) | 50 (40, 57) | 46 (37, 56) | 62 (50, 65) | **0.044** |
| Sex | 189 | | | | | **0.005** |
| Male | | 160 (85) | 126 (83) | 27 (100) | 7 (64) | |
| Female | | 29 (15) | 25 (17) | 0 | 4 (36) | |
| Race/Ethnicity | 189 | | | | | **0.021** |
| White | | 83 (44) | 64 (42) | 15 (56) | 4 (36) | |
| African American | | 59 (31) | 54 (36) | 1 (4) | 4 (36) | |
| Hispanic | | 30 (16) | 21 (14) | 6 (22) | 3 (27) | |
| Other | | 17 (9) | 12 (8) | 5 (18) | 0 | |
| Ever-cigarette smoker | 188 | 122 (65) | 91 (61) | 23 (85) | 8 (73) | **0.039** |
| Ever injection drug use | 186 | 73 (39) | 55 (37) | 13 (50) | 5 (45) | 0.42 |
| **Clinical characteristics** | | | | | | |
| BMI (kg/m$^2$) | 181 | 27.3 ± 6.2 | 27.9 ± 6.5 | 25.9 ± 4.4 | 23.5 ± 4.1 | **0.010** |
| CD4 count (cells/μL) | 184 | 509 (276, 689) | 523 (312, 691) | 413 (170, 571) | 455 (249, 836) | 0.25 |
| Viral load (copies/mL) | 52 | 9761 (464, 60560) | 4142 (251, 44303) | 63400 (23829, 214031) | 103 (103, 103) | **0.022** |
| Undetectable viral load | 131 | 131 (72) | 104 (71) | 17 (68) | 10 (91) | 0.38 |
| Current ART use | 163 | 143 (88) | 114 (87) | 20 (90) | 9 (82) | 0.81 |
| History of TB | 187 | 13 (7) | 11 (7) | 1 (4) | 1 (9) | 0.63 |
| History of PCP | 185 | 43 (23) | 24 (16) | 12 (46) | 7 (64) | **<0.001** |
| History of bacterial pneumonia | 184 | 88 (48) | 66 (45) | 13 (50) | 9 (81) | 0.059 |
| **Symptom data** | | | | | | |
| mMRC | 189 | 1 (0, 1) | 1 (0, 1) | 0 (0, 1.5) | 3 (1, 3) | **0.007** |
| CAT score | 156 | 8 (3, 13) | 7 (2, 12) | 11 (6, 17) | 18 (10, 24) | **0.002** |
| SGRQ score | 189 | 11 (3, 28) | 10 (3, 25) | 20 (11, 28) | 51 (23, 61) | **<0.001** |
| **Pulmonary function testing data** | | | | | | |
| Post- BD FEV1, L | 189 | 3.47 (2.98, 4.21) | 3.46 (2.94, 4.21) | 3.74 (3.42, 4.36) | 2.84 (2.12, 3.09) | **0.002** |
| Post- BD FEV1, % predicted | 189 | 100.7 (92.0, 113.3) | 102.5 (93.5, 115.2) | 94.9 (88.9, 105.1) | 95.9 (93.0, 105.1) | **0.023** |
| Post- BD FVC, L | 189 | 4.31 (3.72, 5.27) | 4.28 (3.70, 5.27) | 4.75 (4.32, 5.65) | 3.71 (2.89, 3.80) | **<0.001** |
| Post- BD FVC, % predicted | 189 | 98.0 (90.3, 110.5) | 99.1 (90.9, 112.1) | 96.1 (90.2, 104.2) | 91.3 (85.1, 117.7) | 0.11 |
| Post- BD FEV1/FVC | 189 | 0.81 (0.77, 0.85) | 0.81 (0.77, 0.85) | 0.77 (0.74, 0.82) | 0.84 (0.78, 0.84) | 0.09 |
| DL$_{CO}$, ml/min/mmHg | 189 | 24.8 (21.0, 28.5) | 26.0 (21.3, 29.5) | 22.6 (20.5, 24.9) | 13.7 (11.3, 16.2) | - |
| DL$_{CO}$, % predicted | 189 | 81.1 (69.8, 89.3) | 84.0 (76.2, 91.1) | 65.7 (63.4, 68.9) | 54.9 (52.6, 56.4) | - |

*Only 52 observations had a detectable viral load

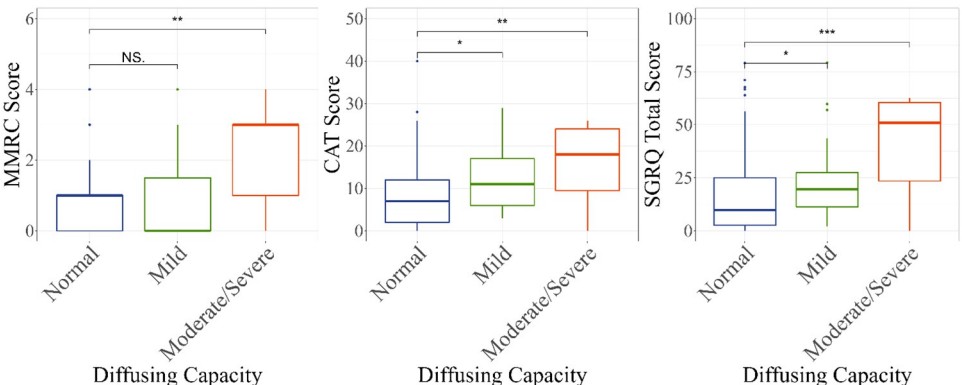

**Fig 2. Unadjusted associations of diffusing capacity for carbon monoxide and respiratory symptom scores** [*P* < **0.05** (*); *P*<**0.01** (**); *P* < **0.001** (***)].

variables significantly associated with greater respiratory symptom scores in our multivariable analyses were higher BMI and history of bacterial pneumonia for all three symptom questionnaires and site location in the mMRC and SGRQ models (higher scores in the San Francisco cohort compared to Seattle cohort, S1 Table). For CAT and SGRQ, history of smoking was significantly associated with abnormal DLco, as was history of injection drug for SGRQ only.

## Iso↓DLco and individual CAT symptom scores

For our secondary outcomes, we tested for associations of iso↓DLco with individual components of the CAT questionnaire. Using a Mantel-Haenszel test for trend, we found that worsening DLco was most strongly associated with increasing scores for breathlessness, activity impairment, and lower energy ($P_{trend}$<0.001; Fig 3A). DLco impairment was also significantly associated with increased cough and with decreased confidence in leaving home ($P_{trend}$ <0.05 for all). We also performed multivariable analyses of individual CAT symptoms. While mild iso↓DLco was only significantly associated with lower energy (aOR 2.48, 95% CI 1.02, 6.12; *P* = 0.048; Table 3), moderate-severe iso↓DLco was significantly associated with breathlessness (aOR 14.9, 95% CI 3.19, 65.8; *P*<0.001), decreased activity (aOR 10.9, 95% CI 3.24, 37.6; *P*<0.001), lower confidence in leaving home (aOR 9.51, 95% CI 2.04, 44.1; *P*<0.001), and lower energy (aOR 9.30, 95% CI 2.82, 32.1; *P*<0.001).

**Table 2. Results from multivariable analyses of mMRC scores, grouped as mMRC <2 and mMRC≥2, CAT, and SGRQ.**

| Multivariable Analysis of Symptom Scores | | | | | | |
|---|---|---|---|---|---|---|
| | Mild iso↓DLco (LLN > DLco > 60% predicted) | | Moderate-severe iso↓DLco (DLco ≤ 60% predicted) | | DLco, % predicted per 1% decrease | |
| | Adjusted Estimate* (95% CI) | *P-value* | Adjusted Estimate (95% CI) | *P-value* | Adjusted Estimate (95% CI) | *P-value* |
| **mMRC (N = 180), ≥ 2 vs <2**[a] | 1.58 (0.54, 4.64) | 0.40 | 7.03 (1.73, 28.5) | **0.006** | 1.05 (1.01, 1.08) | **0.004** |
| **CAT Scores (N = 145)**[b] | 2.85 (-0.18, 5.89) | 0.065 | 9.44 (2.34, 16.5) | **0.009** | 0.15 (0.06, 0.23) | **0.001** |
| **SGRQ Scores (N = 176)**[c] | 9.29 (0.04, 18.5) | **0.049** | 26.7 (4.96, 48.4) | **0.016** | 0.41 (0.22, 0.61) | **<0.001** |

*The adjusted estimates represent odds ratios for mMRC and the predicted mean increase in symptoms score for CAT and SGRQ.

[a]Model adjusted for age, sex, BMI, smoking status, history of bacterial pneumonia, and cohort (Seattle or San Francisco)

[b]Model adjusted for age, sex, BMI, smoking status, and history of bacterial pneumonia

[c]Model adjusted for age, sex, BMI, smoking status, ever injection drug use, history of bacterial pneumonia, and cohort (Seattle or San Francisco)

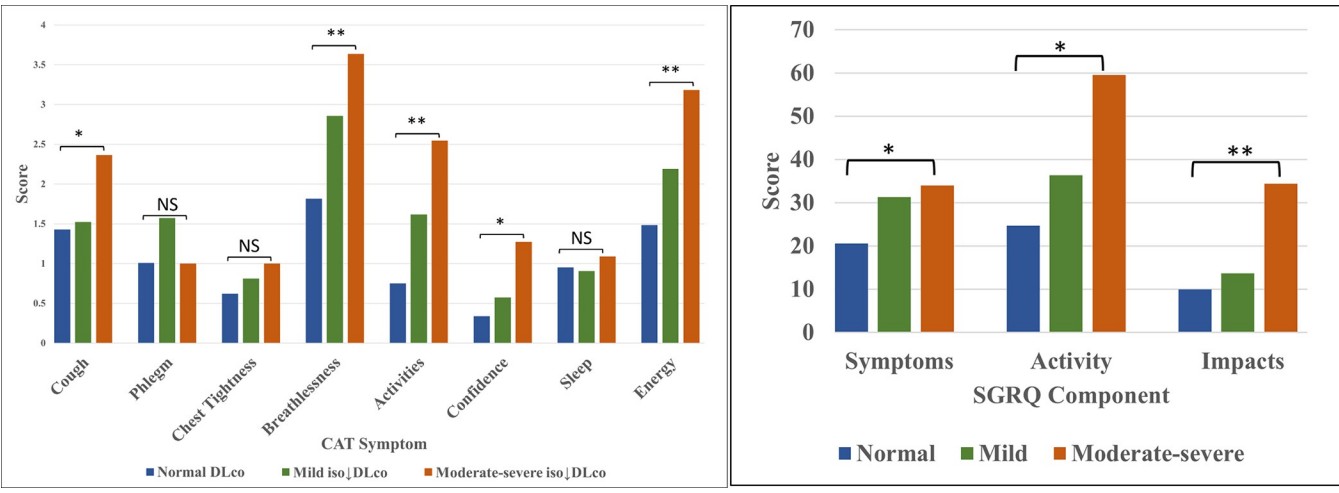

**Fig 3. a.** Distribution of CAT scores by DLco category. Mantel-Haenszel test was used to test for a linear trend between CAT scores and DLco category [$P_{trend}$ < 0.05 (*); $P_{trend}$ < 0.001 (**)]. **b.** Distribution of SGRQ scores by DLco category. Kruskal-Wallis test was used to test for a linear trend between SGRQ scores and $DL_{CO}$ category [$P_{trend}$ < 0.05 (*); $P_{trend}$ < 0.001 (**)].

### Iso↓DLco and SGRQ component scores

Using a Kruskal-Wallis test for trend, we studied the association of iso↓DLco with the SGRQ component scores–symptoms, activity, and impacts—and found that each component score increased significantly as DLco impairment became more severe ($P_{trend}$ <0.001 for impacts, $P_{trend}$ <0.05 for symptoms and activity; Fig 3B).

### Symptom scores with DLco as a continuous measurement

Finally, we performed multivariable analyses of our symptom scores using DLco as a continuous variable in order to ensure that we have not missed any important trends when using iso-↓DLco as a categorical variable. We found that for mMRC, each 1% decrease in % predicted DLco increased the odds of having an mMRC≥2 by 5% (aOR 1.05, 95% CI 1.01, 1.08; $P$ = 0.004; Table 2). Further, we found that for each 1% decrease in DLco, the total CAT score increased by 0.15 points (95% CI 0.06, 0.23; $P$ = 0.001), and the SGRQ score increased by 0.41 points (95% CI 0.22, 0.61; $P$<0.001). We also repeated our multivariable analyses using each individual CAT question and, consistent with our prior analyses, found a significant association between DLco decrease and breathlessness (aOR 1.05, 95% CI 1.03,1.08; $P$<0.001; Table 3), activities (aOR 1.05, 95% CI 1.03, 1.08; $P$<0.001), and energy (aOR 1.04, 95% CI 1.02, 1.06; $P$ = 0.001).

### Discussion

In this study of PWH with normal spirometry, abnormal DLco was significantly associated with a greater respiratory symptom burden and worse quality of life. This relationship strengthened as the degree of DLco impairment increased. Worsening diffusion impairment was also specifically associated with increased dyspnea, lower energy, decreased activity, and lower confidence in leaving home. Together, these findings suggest that the iso↓DLco lung function pattern has clinically relevant symptom correlates and warrants further study. Given that abnormal DLco is the most common PFT abnormality among PWH [4, 7, 9, 35], understanding the etiology of iso↓DLco and devising effective treatment strategies has the potential to improve the quality of life of many PWH.

**Table 3. Results from multivariable analyses of individual CAT questions.**

| | Multivariable Analysis of Symptom Scores | | | | | |
|---|---|---|---|---|---|---|
| | Mild iso↓DLco (LLN > DLco > 60% predicted) | | Moderate-severe iso↓DLco (DLco ≤ 60% predicted) | | DLco, % predicted per 1% decrease | |
| Individual CAT Questions[a] | aOR* (95% CI) | P-value | aOR (95% CI) | P-value | aOR (95% CI) | P-value |
| Cough | 1.37 (0.57, 2.32) | 0.48 | 2.96 (0.89, 10.2) | 0.08 | 1.02 (0.99, 1.04) | 0.055 |
| Phlegm | 2.21 (0.88, 5.55) | 0.09 | 1.35 (0.35, 4.93) | 0.65 | 1.02 (0.99, 1.04) | *0.21* |
| Chest Tightness | 0.92 (0.29, 2.64) | 0.89 | 2.25 (0.58, 8.17) | 0.22 | 1.02 (0.99, 1.04) | *0.24* |
| Breathlessness | 2.14 (0.92, 5.02) | 0.08 | 14.9 (3.19, 65.8) | <**0.001** | 1.05 (1.03, 1.08) | <**0.001** |
| Activities | 2.61 (0.94, 3.23) | 0.06 | 10.9 (3.24, 37.6) | <**0.001** | 1.05 (1.03, 1.08) | <**0.001** |
| Confidence | 2.20 (0.52, 8.12) | 0.25 | 9.51 (2.04, 44.1) | <**0.001** | 1.03 (0.99, 1.08) | 0.08 |
| Sleep | 1.28 (0.45, 3.48) | 0.62 | 1.62 (0.42, 5.71) | 0.46 | 1.01 (0.99, 1.04) | 0.28 |
| Energy | 2.48 (1.02, 6.12) | **0.048** | 9.30 (2.82, 32.1) | <**0.001** | 1.04 (1.02, 1.06) | **0.001** |

*aOR: adjusted odds ratios.

[a]Model adjusted for age, sex, BMI, smoking status, and history of bacterial pneumonia

HIV infection was shown to be an independent risk factor for diffusion abnormalities in several multicenter cohort studies [4, 7, 8, 20], and the burden of diffusion impairment in PWH is significantly higher than in HIV-negative individuals [4, 7]. In PWH, abnormal diffusing capacity regardless of spirometry is associated with increased cough, phlegm, and dyspnea (mMRC ≥2) [4, 7], a shorter six-minute walk distance [36], and increased mortality [14, 37]. In these studies, however, the independent contribution of abnormal diffusion versus the contribution from abnormal spirometry cannot be ascertained. Our study has filled this knowledge gap, and we found that iso↓DLco demonstrated a strong association with respiratory symptoms in the absence of spirometric obstruction or restriction. Our results were more significant in moderate-severe iso↓DLco compared to mild iso↓DLco, and the findings were consistent when DLco was analyzed as a continuous variable. Unsurprisingly, moderate-severe iso↓DLco was most strongly associated with increased breathlessness, although other related symptoms such as activity, energy, and confidence leaving the home also showed significant associations. These results are consistent from a mechanistic perspective as isolated diffusion impairment should cause breathlessness due to gas exchange limitation rather than chronic cough or mucus production as is seen in patients with obstructive or restrictive lung disease. Taken together, these findings suggest that diffusion impairment alone can significantly contribute to symptom burden, particularly breathlessness.

Prior studies have found that DLco impairment (with or without normal spirometry) among PWH is associated with higher HIV RNA levels, CD4<200 cells/μl, and lower nadir CD4 cell count, as well as a history of bacterial pneumonia, *Pneumocystis* pneumonia, cocaine use, and positive hepatitis C RNA [4, 7–9, 38–42]. Abnormal DLco (regardless of spirometry) has been additionally associated with increased markers of inflammation, including elevated serum IL-6 levels [21], as well as higher levels of PARC/CCL-18 and CC-16 pneumoproteins [43]. We similarly found that iso↓DLco groups had higher rates of prior PCP pneumonia and tobacco use compared to the normal PFT group, and the mild iso↓DLco group had higher HIV RNA levels. Interestingly, the moderate-severe iso↓DLco group had a lower HIV viral load, suggesting better virologic control in this group. While the mechanisms that underlie iso↓DLco are unclear, our findings align with prior observations that abnormal DLco is associated with higher levels of immune activation and systemic inflammation.

Iso↓DLco in PWH has a distinct biomarker signature. A prior study from our group examined the associations between 12 plasma biomarkers of immune activation and inflammation

and lung function abnormalities and found that the biomarkers associated with iso↓DLco largely differ from those associated with spirometric obstruction or combined obstruction and diffusion impairment [10]. Specifically, iso↓DLco was associated with sCD14, sCD163, IP10, sTNFR-I and sTNFR-II; spirometric obstruction alone was associated with higher levels of IL-6, and abnormal DLco with spirometric obstruction was associated with IP10, IL-6, fibrinogen, and sTNFR-II. The unique pattern of immune activation in iso↓DLco suggests that there may be a distinct mechanistic pathway driving the development of iso↓DLco among PWH, independent from the pathways leading to COPD/emphysema.

Given that iso↓DLco has a higher prevalence among PWH and demonstrates a unique biomarker signature, iso↓DLco may represent a unique HIV pulmonary disease process. However, drivers of iso↓DLco are likely multifactorial and may include early emphysema, interstitial lung disease, or pulmonary hypertension [44, 45]. Our DLco values were corrected for hemoglobin and carboxyhemoglobin, excluding anemia and methemoglobinemia as potential explanations. At an early stage, emphysema and interstitial lung disease might result in mild reductions in DLco without spirometric abnormalities, though we would expect that moderate-to-severe diffusion impairment would be accompanied by at least mildly abnormal spirometry. Correlation with high-resolution computed tomography (CT) of the chest and echocardiography, which will allow us to assess metrics such as % interstitial lung abnormalities, % emphysema, pulmonary artery systolic pressure, and left ventricular ejection fraction, as well as longitudinal lung function testing and additional testing of inflammatory and immune activation biomarkers are currently being collected and will help address this question.

Our study's main strength is that we used data from a well-established, well-characterized, and diverse multicenter cohort. It builds upon our prior finding that iso↓DLco is associated with a unique set of inflammatory markers and provides a clinical context to these observations. Our limitations include a relatively small sample size, which may have decreased our statistical power to detect significant associations, and male predominance, which decreased the generalizability to women with HIV. In addition, unlike other cohorts of PWH, ours has a higher prevalence of history of pneumonia because of our recruitment strategy [10]. Pneumonia in PWH can cause persistent PFT abnormalities [38], and while previous studies have reported both significantly lower [8] and higher [7] share of overall DLco abnormalities based on the risk profile of each cohort, we may have a higher residual symptom burden in individuals with recent history of pneumonia. Without an HIV-negative comparison group, we cannot definitively assert that this phenomenon is unique to PWH. Finally, lack of imaging and echocardiography data preclude us from determining whether iso↓DLco is a marker for subclinical obstructive, restrictive, or cardiovascular disease, or a separate entity. Further studies are planned to determine the etiology of iso↓DLco and address identified limitations.

## Conclusion

PWH with normal spirometry and abnormal diffusing capacity (iso↓DLco) have a greater respiratory symptom burden based on respiratory symptom questionnaire scores, and symptom scores increased with worsening iso↓DLco. Moderate-severe iso↓DLco was most significantly associated with breathlessness, lower activity, energy, and confidence leaving home. These findings indicate that moderate-severe iso↓DLco is a clinically important PFT finding, and further studies of its underlying mechanism are needed to guide therapeutic interventions.

## Supporting information

**S1 File. Minimal required data set.**
(CSV)

**S1 Table. Outcomes of multivariable analyses of predictors associated with abnormal DLco.**
(DOCX)

## Acknowledgments

The authors gratefully acknowledge the contributions of Wayne Weeks, Eula Lewis, Serena Fong, Stephen Stone, Matthew Sommers, Maria Tercero Paz, John Zoscak, Diane Jeon, Giuliana Lee, Alexander Wong, Charles Lin, Shahida Shahrir, Laurie Hogl, and James Kashima to this study.

## Author Contributions

**Conceptualization:** Katerina L. Byanova, Peter W. Hunt, Kristina A. Crothers, Laurence Huang.

**Data curation:** Jessica Fitzpatrick, Jenny Hsieh.

**Formal analysis:** Jessica Fitzpatrick.

**Funding acquisition:** Peter W. Hunt, Kristina A. Crothers, Laurence Huang.

**Investigation:** Amanda K. Jan, Maggie McGing, Marlena Hartman-Filson, Carly K. Farr, Michelle Zhang, Kendall Gardner, Jake Branchini, Robert Kerruish, Sharvari Bhide, Aryana Bates, Jenny Hsieh, Laurence Huang.

**Methodology:** Katerina L. Byanova, Jessica Fitzpatrick, Kristina A. Crothers, Laurence Huang.

**Project administration:** Laurence Huang.

**Resources:** Amanda K. Jan, Maggie McGing, Marlena Hartman-Filson, Carly K. Farr, Michelle Zhang, Kendall Gardner, Jake Branchini, Robert Kerruish, Sharvari Bhide, Aryana Bates.

**Supervision:** Katerina L. Byanova, Laurence Huang.

**Validation:** Rebecca Abelman, Richard J. Wang.

**Visualization:** Katerina L. Byanova, Jessica Fitzpatrick.

**Writing – original draft:** Katerina L. Byanova.

**Writing – review & editing:** Katerina L. Byanova, Jessica Fitzpatrick, Amanda K. Jan, Maggie McGing, Marlena Hartman-Filson, Carly K. Farr, Michelle Zhang, Kendall Gardner, Jake Branchini, Robert Kerruish, Sharvari Bhide, Aryana Bates, Jenny Hsieh, Rebecca Abelman, Peter W. Hunt, Richard J. Wang, Kristina A. Crothers, Laurence Huang.

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
