## [Decision Letter · Decision Letter 0]

27 Apr 2023

PONE-D-23-02834Isolated abnormal diffusing capacity for carbon monoxide (iso↓DLco) is associated with increased respiratory symptom burden in people with HIVPLOS ONE

Dear Dr. Byanova,

Thank you for submitting your manuscript to PLOS ONE. After careful consideration, we feel that it has merit but does not fully meet PLOS ONE’s publication criteria as it currently stands. Therefore, we invite you to submit a revised version of the manuscript that addresses the points raised during the review process. You may consider revising your manuscript in lines with the comments/suggestions of the reviewers given below. Although there are no major reservations expressed by the reviewers kindly ensure to address all the comments/suggestions with appropriate responses to enable us to further consider your manuscript for publication. Also kindly modify the title of your manuscript to '*Isolated abnormal diffusing capacity for carbon monoxide (iso↓DLco) is associated with increased respiratory symptom burden in people with HIV **infection***' for better clarity.

We look forward to receiving your revised manuscript.

Kind regards,

Koustubh Panda, M. Tech., Ph.D

Academic Editor

PLOS ONE

Journal Requirements:

3. We note that you have stated that you will provide repository information for your data at acceptance. Should your manuscript be accepted for publication, we will hold it until you provide the relevant accession numbers or DOIs necessary to access your data. If you wish to make changes to your Data Availability statement, please describe these changes in your cover letter and we will update your Data Availability statement to reflect the information you provide

Additional Editor Comments:

Kindly modify the title to 'Isolated abnormal diffusing capacity for carbon monoxide (iso↓DLco) is associated with increased respiratory symptom burden in people with HIV **infection**'

Reviewers' comments:

Reviewer's Responses to Questions

**Comments to the Author**

1. Is the manuscript technically sound, and do the data support the conclusions?

Reviewer #1: Yes

Reviewer #2: Yes

2. Has the statistical analysis been performed appropriately and rigorously? 

Reviewer #1: Yes

Reviewer #2: Yes

3. Have the authors made all data underlying the findings in their manuscript fully available?

Reviewer #1: Yes

Reviewer #2: Yes

4. Is the manuscript presented in an intelligible fashion and written in standard English?

Reviewer #1: Yes

Reviewer #2: Yes

5. Review Comments to the Author

Reviewer #1: line 98: Ref 12 this ref shows that the preponderance of pts with decreased DLCO had combined emphysema/ILD causing presumed pseudonormalization of PFTs. This should be mentioned.

line 121: Refs 12 & 20 indicate that decreased DLCO is associated with respiratory symptoms. I would delete this line as it doesn't help.

line 127: Excellent statement of intent!

Materials, Methods, Design, Collection, Analysis are all excellent, as expected from this group.

Table 1: while not the outcome, a plot of viral load against DLCO (whether % predicted or absolute or corrected is up to you from a data vis perspective) would be helpful. I wonder if this ends up a ^-shaped curve since the VL is low in normal DLCO, high in mild DLCO, and low in severe DLCO. If you do this, it will have to be discussed in the text.

Table 2: Can there be a model "d" that includes accounting for viral load?

line 326: This analysis feels like an afterthought, and is hard for many to understand (% predicted per 1% decrease), but DLCO is a continuous variable and a continuous analysis rather than categorical is very important.

line 326-335: can we get a plot of DLCO against symptoms to perhaps identify that there is a different threshold that we should use other than 60%?

line 341: "Together, these findings suggest that isoVDLCO lung function is a distinct condition that has clinically relevant symptom correlates...." Since this isn't a genetic study, the use of phenotype is misleading and too casual a use of the term phenotype.

line 374: You should note that the mod-severe DLCO group had a lower viral load. It is counterintuitive, but an important outlier from the "expected" outcomes.

line 390 : replace "function phenotype" with "disease process"

line 397-399: should you put in presumed measures like ILD score, emphysema scores, RVSP, and evolution of restriction and obstruction?

line 420, right before the last word "these", perhaps insert "These findings are borne out in analysis of DLCO as a continuous variable and suggest a severity cutoff for clinical use of <40%, 50%, 60%, whatever the plot shows>"

Reviewer #2: -This is a well written manuscript and topic is very relevant.

- Can you please specify whether an echo was done to further assess isolated reduction in diffusion capacity. As HIV is a risk factor for pulmonary hypertension which can cause reduced diffucion capacity with relatively normal lungs volumes.

- Did patients go through chest imaging to further asses the cause of low diffusion capacity. Combined pulmonary fibrosis and emphysema (CPFE) can also give pseudo-normal PFTs with significant reduction in diffusion capacity.

-HIV is also associated with heart disease. And patients with chronic heart failure may also have low diffusion capacity. Can you please specify whether patients were screened for signs of heart failure.

6. PLOS authors have the option to publish the peer review history of their article (what does this mean?). If published, this will include your full peer review and any attached files.

Reviewer #1: **Yes: **Arjun B. "Raja" Chatterjee, MD, MS

Reviewer #2: No

---

## [Author Response · Author response to Decision Letter 0]

8 Jun 2023

Please refer to uploaded "Response to Reviewers" document for a copy of these comments and formatted tables. 

Re: PONE-D-23-02834, “Isolated abnormal diffusing capacity for carbon monoxide (iso↓DLco) is associated with increased respiratory symptom burden in people with HIV”

Dear Dr. Panda and PLOS ONE reviewers:

Thank you for your review and insightful feedback on our manuscript “Isolated abnormal diffusing capacity for carbon monoxide (iso↓DLco) is associated with increased respiratory symptom burden in people with HIV.” Please see below for our point-by-point response to your comments and recommendations.

Responses to the editor’s comments:

1. Kindly modify the title to 'Isolated abnormal diffusing capacity for carbon monoxide (iso↓DLco) is associated with increased respiratory symptom burden in people with HIV infection'

Response: We have modified the title of our manuscript in accordance with your recommendation:

 “Isolated abnormal diffusing capacity for carbon monoxide (iso↓DLco) is associated with increased respiratory symptom burden in people with HIV infection”.

Responses to reviewers' comments:

Reviewer #1:

1. Line 98: Ref 12 this ref shows that the preponderance of pts with decreased DLCO had combined emphysema/ILD causing presumed pseudonormalization of PFTs. This should be mentioned.

Response: Thank you for highlighting this important point. We have added a sentence to the paragraph to explain the findings in Aduen et al,

“One study specifically noted that in their single-center experience, iso↓DLco was a result of pseudonormalization of spirometry due to the presence of combined emphysema and restrictive lung disease[12].”

2. Line 121: Refs 12 & 20 indicate that decreased DLCO is associated with respiratory symptoms. I would delete this line as it doesn't help.

Response: We were unable to find reporting of an association between DLco and respiratory symptoms in Raju et al’s manuscript (reference 20). While you are correct that Aduen et al does evaluate based on respiratory symptoms, our original intent was to say that among people with HIV, this issue has not been specifically addressed. In order to improve clarity, we have amended our statement to say,

“Whether iso↓DLco among PWH is associated with any respiratory symptoms or whether it is simply a marker of future complications is not currently known.”

3. Line 127: Excellent statement of intent!

Response: Thank you.

4. Materials, Methods, Design, Collection, Analysis are all excellent, as expected from this group. Table 1: while not the outcome, a plot of viral load against DLCO (whether % predicted or absolute or corrected is up to you from a data vis perspective) would be helpful. I wonder if this ends up a ^-shaped curve since the VL is low in normal DLCO, high in mild DLCO, and low in severe DLCO. If you do this, it will have to be discussed in the text.

Response: This is an excellent suggestion. We have produced two figures showing the distribution of viral load across measured and %predicted DLco (see below). While you are correct that most people with high viral loads are in the mild iso↓DLco group, there are also a lot of individuals with low or suppressed viral RNA in the same group, suggesting that overall, there is no clear relationship between viremia and abnormal DLco in this group. Therefore, we do not think that the pattern is significant enough to warrant the addition of a figure or further discussion in the text.

5. Table 2: Can there be a model "d" that includes accounting for viral load?

Response: Thank you for this excellent suggestion. We did not initially include viral load in our models based on P>0.2 for viral load in our bivariate analyses. Per your recommendation, we performed a sensitivity analysis, in which we added viral load as a covariate (coded as a binary variable, ‘detectable’ vs ‘undetectable’). Please see below the results for table A (equivalent to table 2 in manuscript) and table B (equivalent to table 3 in manuscript) after addition of viral load. In the first table, the only difference in statistical significance between the original analysis and this one is that the association between SGRQ and mild iso↓DLco was attenuated after addition of viral load (p=0.079). In table 3, the association between cough and DLco %predicted newly reached statistical significance with p=0.048 in the sensitivity analysis. Because these results do not substantively change our interpretation of the results, we have chosen to keep the original analysis in our paper.

Associations of iso↓DLco and symptom scores after additional adjustment for viral load.

Table A: Results from multivariable analyses of mMRC scores, grouped as mMRC <2 and mMRC≥2, CAT, and SGRQ. 

Multivariable Analysis of Symptom Scores

 Mild iso↓DLco

(LLN > DLco > 60% predicted) Moderate-severe iso↓DLco 

(DLco ≤ 60% predicted) DLCO, % predicted per 1 % decrease

 Adjusted Estimate (95% CI) P-value Adjusted Estimate (95% CI) P-value Adjusted Estimate (95% CI) P-value

mMRC (N=180), ≥ 2 vs <2a 1.57 (0.54, 4.58) 0.41 6.97 (1.64, 29.5) 0.008 1.05 (1.01, 1.08) 0.004

CAT Scores (N =145)b 2.72 (-0.57, 5.96) 0.10 9.29 (2.40, 16.2) 0.008 0.14 (0.055, 0.22) 0.001

SGRQ Scores (N=176)c 8.53 (-0.99, 18.06) 0.079 25.65 (4.58, 46.78) 0.017 0.41 (0.21, 0.60) <0.001

The adjusted estimates represent odds ratios for mMRC and the predicted mean increase in symptoms score for CAT and SGRQ.

aModel adjusted for age, sex, BMI, smoking status, cohort (Seattle or San Francisco), history of bacterial pneumonia, and detectable HIV viral load

bModel adjusted for age, sex, BMI, smoking status, and history of bacterial pneumonia and detectable HIV viral load 

cModel adjusted for age, sex, BMI, smoking status, ever injection drug use, history of bacterial pneumonia, and cohort (Seattle or San Francisco) and detectable HIV viral load

Table B: Results from multivariable analyses of individual CAT questions.

Multivariable Analysis of Symptom Scores

 Mild iso↓DLco

(LLN > DLco > 60% predicted) Moderate-severe iso↓DLco 

(DLco ≤ 60% predicted) DLCO, % predicted per 1 % decrease

Individual CAT Questionsd Adjusted Estimate (95% CI) P-value Adjusted Estimate (95% CI) P-value Adjusted Estimate (95% CI) P-value

Cough 1.50 (0.61, 3.67) 0.38 3.11 (0.91, 11.0) 0.071 1.02 (1.00, 1.04) 0.048

Phlegm 2.08 (0.80, 5.32) 0.13 1.34 (0.34, 4.95) 0.67 1.01 (0.99, 1.04) 0.37

Chest Tightness 0.99 (0.31, 2.91) 0.99 2.46 (0.63, 8.93) 0.18 1.02 (0.99, 1.04) 0.32

Breathlessness 2.03 (0.85, 4.89) 0.11 14.6 (3.72, 64.6) <0.001 1.05 (1.02, 1.07) <0.001

Activities 2.37 (0.82, 6.65) 0.10 10.30 (3.06, 35.5) <0.001 1.05 (1.03, 1.08) <0.001

Confidence 1.70 (0.33, 6.96) 0.48 8.99 (1.90, 42.2) 0.0050 1.03 (0.99, 1.07) 0.13

Sleep 1.07 (0.35, 3.03) 0.89 1.62 (0.42, 5.65) 0.47 1.01 (0.98, 1.03) 0.50

Energy 2.38 (0.94, 6.11) 0.067 9.54 (2.83, 33.8) <0.001 1.04 (1.02, 1.06) 0.010

The adjusted estimates represent odds ratios.

dModel adjusted for age, sex, BMI, smoking status, history of bacterial pneumonia, and detectable HIV viral load

6. Line 326: This analysis feels like an afterthought, and is hard for many to understand (% predicted per 1% decrease), but DLCO is a continuous variable and a continuous analysis rather than categorical is very important.

Response: We appreciate this insight. The analysis was performed because we wanted to capture associations and trends that may be lost when using a categorical variable based on ‘severity’ cutoffs. Ultimately, we highlighted the categorical outcomes more than the outcome for continuous DLco %predicted in order to emphasize that the most significant pathology is likely to be found in the moderate-severe iso↓DLco group, and that this group should receive the greatest clinical and scientific attention. For clarity, we expanded the first sentence in this section to reflect our intent in evaluating DLco as a continuous outcome (lines 355-357), and we rephrased lines 357-358 in the revised manuscript to be explicit in the interpretation of the odds ratios: 

“Finally, we performed multivariable analyses of our symptom scores using DLco as a continuous variable in order to ensure that we have not missed any important trends when using iso↓DLco as a categorical variable. We found that for mMRC, each 1% decrease in % predicted DLco increased the odds of having an mMRC≥2 by 5% (aOR 1.05, 95% CI 1.01, 1.08; P=0.004; Table 2).”

7. Line 326-335: can we get a plot of DLCO against symptoms to perhaps identify that there is a different threshold that we should use other than 60%?

Response: Thank you for this excellent suggestion. We have produced scatterplots of CAT and SGRQ scores against DLco % predicted (see below). Based on these plots, we were not able to identify a more suitable cutoff for normal/abnormal DLco. This is likely a reflection of both the multiple possible etiologies that could drive iso↓DLco and respiratory symptoms, as well as of the fact that these questionnaires are not specific for the assessment of iso↓DLco but rather a range of pulmonary pathologies.

8. Line 341: "Together, these findings suggest that isoVDLCO lung function is a distinct condition that has clinically relevant symptom correlates...." Since this isn't a genetic study, the use of phenotype is misleading and too casual a use of the term phenotype.

Response: We understand your point. Our use of the word ‘phenotype’ was based on Merriam-Webster’s dictionary’s definition of the word as “the observable characteristics or traits of a disease.” However, we do recognize that iso↓DLco is a presentation of one or more possible underlying conditions, and, therefore, we have changed the word “phenotype” to “pattern” in the line you pointed out, as well as in line 145 of the revised manuscript (which are the only two occurrences in the text) to keep our language more precise. The sentence in line 374-375 of the revised manuscript now reads,

“Together, these findings suggest that the iso↓DLco lung function pattern has clinically relevant symptom correlates and warrants further study.”

9. Line 374: You should note that the mod-severe DLCO group had a lower viral load. It is counterintuitive, but an important outlier from the "expected" outcomes.

Response: Your point is well taken. We have added a sentence in line 413-415 of the revised manuscript that discusses this issue:

“Interestingly, the moderate-severe iso↓DLco group had a lower HIV viral load, suggesting better virologic control in this group.”

10. Line 390: replace “function phenotype” with “disease process”

Response: This change has been made: 

“Given that iso↓DLco has a higher prevalence among PWH and demonstrates a unique biomarker signature, iso↓DLco may represent a unique HIV pulmonary disease process.”

11. Line 397-399: should you put in presumed measures like ILD score, emphysema scores, RVSP, and evolution of restriction and obstruction?

Response: We have updated the last sentence of the paragraph (revised manuscript lines 437-442) to be explicit about the kinds of measurements we would like to make in our future analyses:

“Correlation with high-resolution computed tomography (CT) of the chest and echocardiography, which will allow us to assess metrics such as % interstitial lung abnormalities, % emphysema, pulmonary artery systolic pressure, and left ventricular ejection fraction, as well as longitudinal lung function testing and additional testing of inflammatory and immune activation biomarkers are currently being collected and will help address this question.”

12. Line 420, right before the last word "these", perhaps insert "These findings are borne out in analysis of DLCO as a continuous variable and suggest a severity cutoff for clinical use of <40%, 50%, 60%, whatever the plot shows>"

Response: Thank you for this excellent point. Since the additional analyses did not show a better cutoff for %predicted DLco, we edited the sentence to highlight that attention should be paid to moderate-severe iso↓DLco abnormalities:

“These findings indicate that moderate-severe iso↓DLco is a clinically important PFT finding, and further studies of its underlying mechanism are needed to guide therapeutic interventions.”

Reviewer #2: 

1. This is a well written manuscript and topic is very relevant. Can you please specify whether an echo was done to further assess isolated reduction in diffusion capacity. As HIV is a risk factor for pulmonary hypertension which can cause reduced diffusion capacity with relatively normal lungs volumes.

Response: Thank you for your comment. We do not have research echocardiograms in our study population, although this is part of our next steps for investigating the etiology of iso↓DLco. We have explained this in our limitations and future studies part of the discussion (lines 437-442):

“Correlation with high-resolution computed tomography (CT) of the chest and echocardiography, which will allow us to assess metrics such as % interstitial lung abnormalities, % emphysema, pulmonary artery systolic pressure, and left ventricular ejection fraction, as well as longitudinal lung function testing and additional testing of inflammatory and immune activation biomarkers are currently being collected and will help address this question.”

Additionally, we did a retrospective chart review for a subset of participants with iso↓DLco who had available clinical echocardiograms in our hospital system, and we found that 15/55 individuals had PASP >30 cm H2O. However, these echocardiograms were not closely matched in time to PFTs and were typically obtained during time of acute illness. Thus, these results were not reliable enough to be included in this manuscript but provide additional support for obtaining TTEs systematically in our cohort.

2. Did patients go through chest imaging to further assess the cause of low diffusion capacity. Combined pulmonary fibrosis and emphysema (CPFE) can also give pseudo-normal PFTs with significant reduction in diffusion capacity.

Response: Thank you again for the comment. This is an excellent point, but similar to above, we did not have chest CTs as part of our research study. We reviewed available clinical CT images from 32 participants and did not find a consistent pattern that was associated with iso↓DLco. Having recognized the lack of imaging data as a limitation, we are currently working on obtaining matched CTs and PFTs for our study participants to address this question.

3. HIV is also associated with heart disease. And patients with chronic heart failure may also have low diffusion capacity. Can you please specify whether patients were screened for signs of heart failure.

Response: Thank you for this comment. Patients were not formally screened for clinical signs of chronic heart failure during study visits, and as explained above, as of now we lack research echocardiography data. However, in our clinical TTE review, we found only one individual who had evidence of left heart disease with an ejection fraction of 45%. We are hoping to address this question in the future by obtaining research TTEs.

We believe that these revisions have strengthened and improved the manuscript and appreciate your helpful suggestions for improvement. We look forward to your reply.

---

## [Decision Letter · Decision Letter 1]

4 Jul 2023

Isolated abnormal diffusing capacity for carbon monoxide (iso↓DLco) is associated with increased respiratory symptom burden in people with HIV infection

PONE-D-23-02834R1

Dear Dr. Byanova,

We’re pleased to inform you that your manuscript has been judged scientifically suitable for publication and will be formally accepted for publication once it meets all outstanding technical requirements.

Kind regards,

Koustubh Panda, M. Tech., Ph.D

Academic Editor

PLOS ONE

Additional Editor Comments (optional):

Reviewers' comments:

Reviewer's Responses to Questions

**Comments to the Author**

1. If the authors have adequately addressed your comments raised in a previous round of review and you feel that this manuscript is now acceptable for publication, you may indicate that here to bypass the “Comments to the Author” section, enter your conflict of interest statement in the “Confidential to Editor” section, and submit your "Accept" recommendation.

Reviewer #2: All comments have been addressed

2. Is the manuscript technically sound, and do the data support the conclusions?

Reviewer #2: Yes

3. Has the statistical analysis been performed appropriately and rigorously? 

Reviewer #2: Yes

4. Have the authors made all data underlying the findings in their manuscript fully available?

Reviewer #2: Yes

5. Is the manuscript presented in an intelligible fashion and written in standard English?

Reviewer #2: Yes

6. Review Comments to the Author

Reviewer #2: My questions in from prior review have been answered. I wish good luck to the authors for continued research work.

7. PLOS authors have the option to publish the peer review history of their article (what does this mean?). If published, this will include your full peer review and any attached files.

Reviewer #2: No

---

## [Editor Report · Acceptance letter]

10 Jul 2023

PONE-D-23-02834R1 

Isolated abnormal diffusing capacity for carbon monoxide (iso↓DLco) is associated with increased respiratory symptom burden in people with HIV infection 

Dear Dr. Byanova:

I'm pleased to inform you that your manuscript has been deemed suitable for publication in PLOS ONE. Congratulations! Your manuscript is now with our production department. 

Kind regards, 

on behalf of

Professor Koustubh Panda 

Academic Editor

PLOS ONE